# Hyper-opinion Evidential Deep Learning for Out-of-Distribution Detection

**Jingen Qu**
School of Computer Science and Technology
Tongji University, Shanghai, China
newcity@tongji.edu.cn

**Yufei Chen**[*]
School of Computer Science and Technology
Tongji University, Shanghai, China
yufeichen@tongji.edu.cn

**Xiaodong Yue**
Artificial Intelligence Institute
Shanghai University, Shanghai, China.
yswantfly@shu.edu.cn

**Wei Fu**
School of Computer Science and Technology
Tongji University, Shanghai, China
cs_fuwei@outlook.com

**Qiguang Huang**
School of Computer Science and Technology
Tongji University, Shanghai, China
1753543@tongji.edu.cn

## Abstract

Evidential Deep Learning (EDL), grounded in Evidence Theory and Subjective Logic (SL), provides a robust framework to estimate uncertainty for out-of-distribution (OOD) detection alongside traditional classification probabilities. However, the EDL framework is constrained by its focus on evidence that supports only single categories, neglecting the other collective evidences that could corroborate multiple in-distribution categories. This limitation leads to a diminished estimation of uncertainty and a subsequent decline in OOD detection performance. Additionally, EDL encounters the vanishing gradient problem within its fully-connected layers, further degrading classification accuracy. To address these issues, we introduce hyper-domain and propose Hyper-opinion Evidential Deep Learning (HEDL). HEDL extends the evidence modeling paradigm by explicitly integrating sharp evidence, which supports a singular category, with vague evidence that accommodates multiple potential categories. Additionally, we propose a novel opinion projection mechanism that translates hyper-opinion into multinomial-opinion, which is then optimized within the EDL framework to ensure precise classification and refined uncertainty estimation. HEDL integrates evidences across various categories to yield a holistic evidentiary foundation for achieving superior OOD detection. Furthermore, our proposed opinion projection method effectively mitigates the vanishing gradient issue, ensuring classification accuracy without additional model complexity. Extensive experiments over many datasets demonstrate our proposed method outperforms existing OOD detection methods.

## 1 Introduction

Deep Learning (DL) models have been widely adopted in many real-world applications[25, 57, 64, 15]. However, these models are trained under the implicit assumption that the training and test data are

---

[*]corresponding author

38th Conference on Neural Information Processing Systems (NeurIPS 2024).

drawn from the same distribution[70], leading to overconfident predictions[45]. Thus when a DL model encounters an input that differs from its training data, it may be overconfident with wrong prediction, bringing rise to the out-of-distribution (OOD) problem. The resolution of the OOD problem is of utmost importance, and researchers have devoted significant attention to studying the intricacies of OOD detection[5, 16, 19, 30, 31, 43].

To address OOD problem, a variety of methods have been developed in DL[12, 4, 51]. Some researchers apply post-processors to the base classifier to generate an uncertainty score for OOD detection. These post-hoc methods only take effect at inference phase and are easy to use, but rely on the performance of the pretrained model. Others propose training methods that involve training-time regularization, which require more computational resources. To train an uncertainty-aware model without additional computation, a recent search leverages Evidence Theory and Subjective Logic (SL) with DNNs[54], called Evidential Deep Learning (EDL)[55, 24, 54, 7]. EDL offers uncertainty estimation in neural networks which represents the degree of 'unknown' in opinion. It modifies the existing DL structure slightly and allows neural network to quantify the uncertainty for OOD detection with a well-defined theory framework. Evidential models have been extended to many areas such as open set recognition[2], classification[35, 32, 22, 36, 33], multi-view learning[72, 68, 23, 34].

The EDL models face several challenges, with one primary issue arising from the theoretical framework. The evidence in multinomial-opinion in EDL exclusively supports singleton sets, which contains only one category. In other words, EDL only captures the evidence which supports single category and rejects others. As a result, EDL is unable to effectively leverage vague evidence, such as features supporting a composite set containing multiple categories. As Figure 1 shows, EDL suffers from performance degradation in the face of ambiguous samples.

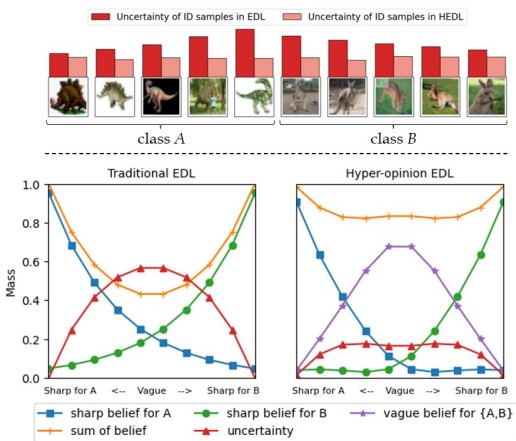

In addition, the parameters of fully-connected layer in EDL models are facing vanishing gradient problem when number of category in datasets rises[49]. Vanishing gradient in EDL leads to failure in classification of several categories. To mitigate this problem, Pandey et al.[49] introduce regularization techniques. However, these efforts yield unsatisfactory results in real world OOD detection tasks.

Figure 1: Belief and uncertainty masses across varying levels of In-distribution sample vagueness. As sample gets vaguer, EDL tends to extract a minimal quantity of sharp evidence, results in elevated uncertainty estimation. HEDL demonstrates the capability to extract vague evidence as sample vagueness increases, thereby maintaining lower uncertainty levels.

To train an evidential model maintaining classification accuracy and providing reliable uncertainty estimation for OOD detection, we incorporate EDL with hyper-opinion and propose Hyper-opinion Evidential Deep Learning (HEDL). While EDL is built upon multinomial-opinion in a basic domain, hyper-opinion represents the opinion in the hyper-domain, which includes the basic domain and the composite sets. Through the concepts of composite set, HEDL is able to learn from vague evidence ignored by EDL. HEDL provides an effective mechanism for quantifying evidence that supports composite sets, thereby enhancing the differentiation of OOD data and classification accuracy. Our major contributions can be summarized as follows:

- We introduce an evidential representation within the hyper-domain, which integrates sharp evidence that supports a singular category, with vague evidence that accommodates multiple potential categories, to establish a more comprehensive and accurate evidentiary foundation.

- We develop a hyper-opinion framework within the hyper-domain and propose a novel opinion projection. This method transfers hyper-opinion to multinomial-opinion, allocating evidence to each category precisely and mitigating the vanishing gradient problem, while preserving computational efficiency.

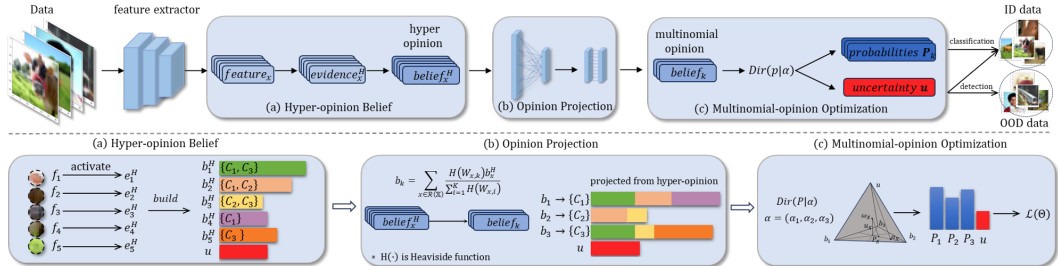

Figure 2: Framework of HEDL. HEDL framework is composed of three integral components. The first part transfers the extracted features to evidence and models them with in hyper-opinion framework. Subsequently, the second component projects the hyper-opinion to multinomial-opinion. Ultimately, the framework optimizes the opinion to attain precise classification and to furnish robust uncertainty estimations for OOD detection.

- Our proposed Hyper-opinion Evidential Deep Learning (HEDL) procures more exhaustive evidence, which refines the precision of uncertainty estimation, and consequently enhances the performance of OOD detection while maintaing ID classification accuracy.

- We carry out experiments over multiple challenging datasets to validate the OOD detection in HEDL outperforms existing OOD detection methods.

## 2 Related Work

### 2.1 Uncertainty based OOD Detection

Accurately quantifying predictive uncertainty in DL models is crucial for recognizing out-of-distribution (OOD) samples. Traditional softmax-based models provide confidence estimation through class posteriors, which are inversely correlated with predictive uncertainty[16]. Several methods applicable to pre-trained classifiers that output class posteriors using softmax have been proposed[3, 14, 53, 37, 60, 18], including Out-of-Distribution Detector for Neural Networks[31] and Mahalanobis Distance[30]. Besides, deep ensemble is a technique developed for uncertainty quantification[29], which constructs an ensemble of neural networks and measures uncertainty based on the agreement/disagreement across the ensemble components[13]. However, this approach significantly increases the scale of model parameters, leading to high computational and storage complexity. Alternatively, neural networks based on Bayesian statistics called Bayesian neural networks[12, 4, 42] is raised to to quantify different uncertainties in Bayesian formalism. Bayesian methods normally apply approximation to address the intractability issue in marginalization of latent variables. And as such methods require sampling for uncertainty quantification, leading to expensive computations. A recent research effort has summarized OOD detection methods and established an OOD benchmark [69].

### 2.2 Evidential Deep Learning

EDL introduces a conjugate higher-order evidential prior for the likelihood distribution that enables the model to capture the evidence vacuity as predictive uncertainty. The training of an EDL model can be regarded as an evidence-collecting process. Researches on multiple applications with EDL have been done, e.g., Dirichlet prior is introduced over the multinomial likelihood for evidential classification[2, 73, 11], evidential models for regression[1, 48], adversarial robustness[27] and calibration[63]. Most existing methods built upon EDL are trained on evidential losses conjunct with regularization of the evidence to guide the evidence vacuity, *i.e.*, uncertainty, behavior[47, 56]. Some EDL models combine with the idea of outlier exposure[17] that provides access of OOD data to guide the evidence learning process of EDL models[40, 41].

In this work, we focus on evidential models for classification and OOD detection, and consider settings where no extra regularization and OOD data are used during model training to make the proposed approach more broadly applicable to practical real-world situations.

# 3 Proposed Method

Our method's framework is depicted in Figure 2, which operates under the assumption of no prior information.

## 3.1 Hyper-opinion Belief

Subjective Logic (SL) is a theory of uncertain reasoning based on probability theory and belief theory in a domain $\mathbb{X}$, which represents the set of exclusive possible states of a variable situation. It introduces the concepts of belief mass and uncertainty mass to describe the degree of belief and uncertainty about an event.

Traditional EDL is built upon multinomial-opinion within domain $\mathbb{X}$ in SL and domain $\mathbb{X}$ is a limited portion of hyper-domain $\mathcal{R}(\mathbb{X})$, where $\mathcal{P}(\mathbb{X})$ is the powerset of $\mathbb{X}$.

$$\mathcal{R}(\mathbb{X}) = \mathcal{P}(\mathbb{X})/\{\{\mathbb{X}\}, \{\emptyset\}\}. \tag{1}$$

Let us consider a domain $\mathbb{X}$ with cardinality of $K$, SL provides a belief mass $b_k$ representing the belief degree and a base rate $a_k$ representing the prior information for each singleton $k = 1, ..., K$ and an overall uncertainty mass of $u$. The three compose a multinomial-opinion $\boldsymbol{\omega} = (\boldsymbol{b}, u, \boldsymbol{a})$, belief mass and uncertainty mass sum up to one, eg.,

$$u + \sum_{k=1}^{K} b_k = 1, \quad u \geq 0 \quad and \quad b_k \geq 0 \quad for \quad k = 1, ..., K. \tag{2}$$

Our method models the evidence in hyper-domain $\mathcal{R}(\mathbb{X})$ with hyper-opinion, which provides a belief mass $b_x^H, x \in \mathcal{R}(\mathbb{X})$, representing the belief degree of set $x$. Along with $\boldsymbol{a}^H$ and $u$, the three compose a hyper-opinion $\boldsymbol{\omega}^H = (\boldsymbol{b}^H, u, \boldsymbol{a}^H)$ and the hypernomial belief mass distribution also follows the additivity requirement:

$$b^H : \mathcal{R}(\mathbb{X}) \to [0, 1]$$
$$u + \sum_{x \in \mathcal{R}(\mathbb{X})} b_x^H = 1. \tag{3}$$

Hyper opinion allows belief mass to be divided into two types called sharp belief mass and vague belief mass. Belief mass that only supports a specific singleton is called sharp belief mass, eg., $k \in \mathbb{X}$, it discriminates between this and other singletons. EDL built upon the multinomial-opinion only offers sharp belief mass estimation. Considers a domain $\mathbb{X}$ of K mutually exclusive singletons, for each singleton $k = 1, ..., K$, sharp belief mass is

$$b_k^{\mathrm{S}} = b_k^H, \forall k \in \mathbb{X}. \tag{4}$$

Belief mass assigned to a composite set $x \in \mathcal{C}(\mathbb{X})$, where $\mathcal{C}(\mathbb{X}) = \mathcal{R}(\mathbb{X})/\mathbb{X}$, represents vague belief mass because it expresses cognitive vagueness. It supports the truth of multiple singletons in $\mathbb{X}$ simultaneously. Vague belief mass can be allocated to a singleton $k$ as

$$b_k^{\mathrm{V}} = \sum_{x \in \mathcal{C}(\mathbb{X})} a(k|x)b_x^H, \quad a(k|x) = \frac{a_k}{\Sigma_{i \in x} a_i}, \quad \forall k \in \mathbb{X}, \forall x \in \mathcal{C}(\mathbb{X}), \tag{5}$$

where $a(k|x)$ is relative base rate. When no prior information is available, $a(k|x)$ can be simplified to

$$a(k|x) = \frac{1}{|x|}, \quad \forall k \in \mathbb{X}, \forall x \in \mathcal{C}(\mathbb{X}), \tag{6}$$

where $|x|$ is the cardinality of $x$. Then in hyper-opinion, a belief mass $b_x^H$ for a set $x$ is computed using the evidence for the set. Let $e_x^H \geq 0$ be the evidence derived for the set $x$, then the belief $b_x^H$ and the uncertainty $u$ are computed as

$$b_x^H = \frac{e_x^H}{S} \quad and \quad u = \frac{KW_{prior}}{S}, \quad S = \sum_{x \in \mathcal{R}(\mathbb{X})} e_x^H + KW_{prior}. \tag{7}$$

By introducing hyper-opinion, vague beliefs that assigned to composite sets can be take into consideration, which better measure comprehensive evidence and estimate uncertainty more accurately.

In practice, we activate the features extracted by the neural network as evidence in hyper-domain, and build them within hyper-opinion to distinguish sharp belief and vague belief. This allows the model to maintain its vagueness among similar in-distribution categories, thereby ensuring that the uncertainty remains low.

## 3.2 Opinion Projection

A projection from hyper-opinion to multinomial-opinion is needed to realize the projected probability of each singleton. Therefore we introduce a novel opinion projection implementation that projects belief mass from hyper-opinion into multinomial-opinion, with $b_k^V$ and $b_k^S$ that can be calculated by Eq. 4 and Eq. 5, following

$$b_k = b_k^V + b_k^S, \forall k \in \mathbb{X}. \tag{8}$$

We activate the features extracted by neural network for ascertaining non-negative evidence within the hyper-domain. After associate evidence with belief in hyper-opinion, we determine the set each belief mass supports as mentioned in section 3.1, and project the belief mass from hyper-opinion to multinomial-opinion.

Specifically, we apply a unit step activation function to the parameters of the fully connected layer, eg., Heaviside function

$$H(x) = \begin{cases} 1, x > 0 \\ 0, else. \end{cases} \tag{9}$$

It offers an access to a matrix $W^S = H(W)$, where $W$ corresponds to the weight matrix of the fully connected layer. $W^S$ represents the information of set each belief mass supports.

Assume there are $K$ singletons and $N$ belief masses supporting different sets, it offers a matrix $W_{N,K}^S$. For a belief mass $b_x^H$ supporting set $x$, $W_x^S$ is a vector that contains information about which singletons belong to the set $x$.

Once the set each belief mass supports has been identified, projecting hyper-opinion to multinomial-opinion is straightforward. For each belief mass within the hyper-opinion, we can compute its relative base rate to each singleton, and allocate belief mass accordingly. For a singleton $k$, its total projected multinomial-opinion belief mass is

$$b_k = \sum_{x \in \mathcal{R}(\mathbb{X}} (b_x^H W_{x,k}^p), \tag{10}$$

$$W_{x,k}^p = \frac{a_k H(W_{x,k})}{\sum_{i=1}^K (a_i H(W_{x,i}))} = \frac{a_k W_{x,k}^S}{\sum_{i=1}^K (a_i W_{x,i}^S)}, \quad k \in \mathbb{X}, x \in \mathcal{R}(\mathbb{X}), \tag{11}$$

where $\boldsymbol{a}$ is the base rate. Without any prior information, Eq. 11 can be simplied to

$$W_{x,k}^p = \frac{W_{x,k}^S}{\sum_{i=1}^K W_{x,i}^S}, \quad k \in \mathbb{X}, x \in \mathcal{R}(\mathbb{X}). \tag{12}$$

To date, we have successfully delineated the process of projecting belief mass from a hyper-opinion to a multinomial-opinion within a neural network framework. In practical terms, this projection is executed by applying a linear transformation to the output of the fully connected layer. This transformation facilitates the allocation of belief mass to the respective singletons in the multinomial-opinion. Consequently, the incremental computational complexity associated with our method is constant as O(1).

$$\boldsymbol{b} = \boldsymbol{o} \cdot G(W, \boldsymbol{b}^H), \quad G(W, \boldsymbol{b}^H) = \frac{W^p \boldsymbol{b}^H}{W \boldsymbol{b}^H}, \tag{13}$$

where $\boldsymbol{o}$ is the output of fully-connected layer and $W, W^p, \boldsymbol{b}^H$ are all detached variables, making $G(W, \boldsymbol{b}^H)$ a constant during one training epoch.

The output after opinion projection represents the projected multinomial-opinion in EDL, which has the equivalent meaning in EDL and can be optimized with the same techniques. We used an example to show why the uncertainty estimation of HEDL outperforms EDL in Appendix A.

## 3.3 Multinomial-opinion Optimization

By building evidence within hyper-domain and projecting hyper-opinion belief mass into multinomial-opinion belief mass, we construct a flow that can be optimized in multinomial-opinion framework to obtain the comprehensive evidence and accurate uncertainty estimation for OOD detection, which is similar to traditional EDL.

As the sum of evidence $\sum_{x \in \mathcal{R}(\mathbb{X})} e_x^H$ and uncertainty $u$ remain the same during the projection, we can pass the belief mass in the form of evidence to simplify the calculation. Therefore the projected probability distribution derived from the projected multinomial-opinion can correspond to an expected probability distribution derived from a Dirichlet distribution parameterized by $\boldsymbol{\alpha}$

$$
\begin{aligned}
\boldsymbol{\omega} &= (\boldsymbol{b}, u, \boldsymbol{a}) \leftrightarrow Dir\left(\boldsymbol{P} \,|\, \boldsymbol{\alpha}\right), \\
\alpha_k &= e_k + a_k W_{prior} = b_k S + a_k W_{prior}.
\end{aligned}
\tag{14}
$$

The Dirichlet distribution is a probability density function (pdf) for possible values of the probability mass function (pmf) $P$ and is given by:

$$
Dir\left(P \,|\, \boldsymbol{\alpha}\right) = \frac{1}{B(\boldsymbol{\alpha})} \prod_i^K p_i^{\alpha_i - 1}.
\tag{15}
$$

In projected multinomial-opinion, the expected probability for the $k^{th}$ singleton calculation is

$$
\hat{p}_k = \frac{\alpha_k}{S},
\tag{16}
$$

which allows to be optimized by the loss function defined in EDL

$$
\mathcal{L}_i(\Theta) = \int \left[ \sum_{j=1}^K -y_{ij} \log(p_{ij}) \right] \frac{1}{B(\alpha_i)} \prod_{j=1}^K p_{ij}^{\alpha_{ij} - 1} d\mathbf{p}_i = \sum_{j=1}^K y_{ij} \Big( \psi(S_i) - \psi(\alpha_{ij}) \Big),
\tag{17}
$$

where $\psi(\cdot)$ is the digamma function, $y_i$ is a one-hot vector encoding the ground-truth class of observation $x_i$ with $y_{ij} = 1$ and $y_{ik} = 0$ for all $k \neq j$, and $\alpha_i$ be the parameters of the Dirichlet density on the predictors.

At this point, we have established the complete framework of HEDL, spanning all stages ranging from input processing to classification and uncertainty estimation. Our method objective has the following proposition in the Appendix B.

By establishing the framework of HEDL, we comprehensively extract the sharp and vague evidence each sample contains and allocate preciously, thereby enabling accurate classification. Moreover, comprehensive evidence contributes to improved uncertainty estimation and subsequently enhances the performance of OOD detection.

## 4 Experiment

In this section, we describe our experimental setup and demonstrate the effectiveness of our method on a wide range of OOD evaluation benchmarks and the most widely used metric AUROC is adopted[52, 21, 10, 37]. We also conduct an ablation analysis that leads to an improved understanding of our approach.

### 4.1 Setup

**In-distribution Datasets.** We use the CIFAR-10[28], CIFAR-100[28], Flower-102[46] and CUB-200-2011[65] as ID data.

**Out-of-distribution Datasets.** For the OOD test datasets, we use three common benchmarks[69]: SVHN[44], Textures[6], Places365[74], that are used in Openood-benchmark[69]. There is no overlapping between ID datasets and OOD datasets.

**Evaluation Metrics.** We measure the following metrics: 1) FPR95 measures the false positive rate (FPR) when the true positive rate (TPR) is equal to 95%. Lower scores indicate better performance. 2) AUROC measures the area under the Receiver Operating Characteristic (ROC) curve, which displays the relationship between TPR and FPR. The area under the ROC curve can be interpreted as the probability that a positive ID example will have a higher detection score than a negative OOD example. 3) AUPR measures the area under the Precision-Recall (PR) curve. The PR curve is created by plotting precision versus recall. AUROC is the most common metric[52, 21, 10, 37] and we use AUROC as the main metric for OOD detection performance while accuracy measures performance of detecting ID samples. Our goal is to detect more OOD samples while maintaining ID classification performance.

Table 1: Comparison of OOD detection performance between HEDL and other baselines with CIFAR-10 and CIFAR-100 as ID dataset. All values are percentages. ↑ indicates larger values are better, and ↓ indicates smaller values are better. The **bold** are superior results.

| Method | SVHN | | | Textures | | | Place365 | | | Average | | | ID data |
|---|---|---|---|---|---|---|---|---|---|---|---|---|---|
| | FPR95↓ | AUPR↑ | AUROC↑ | FPR95↓ | AUPR↑ | AUROC↑ | FPR95↓ | AUPR↑ | AUROC↑ | FPR95↓ | AUPR↑ | AUROC↑ | Acc.↑ |
| **CIFAR-10** | | | | | | | | | | | | | |
| MSP[16] | 51.87 | 78.19 | 90.88 | 59.89 | 91.28 | 88.72 | 57.64 | 70.24 | 89.03 | 56.47 | 79.90 | 89.54 | 95.06 |
| ODIN[31] | 67.92 | 42.13 | 73.32 | 51.10 | 82.25 | 80.70 | 50.51 | 50.27 | 82.55 | 56.51 | 58.22 | 78.86 | 95.06 |
| openGAN[26] | 99.39 | 33.90 | 53.56 | 98.24 | 61.48 | 42.22 | 99.44 | 19.55 | 36.58 | 99.02 | 38.31 | 44.12 | 95.06 |
| GradNorm[21] | 91.65 | 78.89 | 53.91 | 98.09 | 48.05 | 52.07 | 92.46 | 86.63 | 60.50 | 94.07 | 71.19 | 55.49 | 95.06 |
| VIM[66] | 14.41 | 93.76 | 97.22 | 20.78 | 97.36 | 96.06 | 47.52 | 72.83 | 90.08 | 27.57 | 87.98 | 94.46 | 95.06 |
| KNN[61] | 33.32 | 92.31 | 95.13 | 46.01 | 95.93 | 92.77 | 43.78 | 80.15 | 91.82 | 41.04 | 89.47 | 93.23 | 95.06 |
| DICE[59] | 67.78 | 73.19 | 86.43 | 67.48 | 85.38 | 80.14 | 56.06 | 57.52 | 84.43 | 63.78 | 72.03 | 83.66 | 95.06 |
| RankFeat[58] | 64.49 | 80.33 | 68.15 | 59.71 | 55.39 | 73.46 | 43.70 | 94.66 | 85.99 | 55.97 | 76.79 | 75.87 | 95.06 |
| ASH[8] | 83.64 | 89.06 | 73.46 | 84.59 | 72.85 | 77.45 | 77.89 | 94.04 | 79.89 | 82.04 | 85.32 | 76.93 | 95.06 |
| SHE[71] | 62.74 | 94.46 | 86.38 | 84.60 | 77.28 | 81.57 | 76.36 | 94.88 | 82.89 | 74.57 | 88.87 | 83.61 | 95.06 |
| GEN[38] | 28.14 | 96.37 | 91.97 | 40.74 | 84.71 | 90.14 | 47.03 | 96.67 | 89.46 | 38.64 | 92.58 | 90.52 | 95.06 |
| MCDropout[12] | 44.58 | 85.03 | 92.67 | 56.60 | 91.74 | 88.83 | 56.20 | 67.20 | 88.43 | 52.47 | 81.32 | 89.98 | 94.95 |
| G-ODIN[19] | 8.42 | 96.63 | 98.41 | 23.32 | 96.03 | 94.51 | 39.80 | 75.49 | 91.10 | 23.84 | 89.39 | 94.67 | 94.70 |
| CSI[62] | 17.56 | **97.75** | 95.18 | 28.95 | 82.99 | 90.71 | 34.76 | **96.38** | 89.56 | 27.09 | 92.37 | 91.82 | 91.16 |
| MOS[20] | 90.85 | 70.55 | 51.09 | 85.56 | 90.89 | 52.91 | 71.74 | 78.67 | 74.15 | 82.71 | 80.03 | 59.38 | 94.83 |
| VOS[9] | 29.92 | 83.73 | 93.82 | 37.38 | 92.72 | 91.26 | 45.37 | 63.93 | 88.73 | 37.55 | 80.13 | 91.27 | **95.82** |
| LogitNorm[67] | **5.30** | 97.70 | **98.86** | 30.94 | 96.32 | 94.30 | 31.17 | 88.11 | 94.76 | 22.47 | 94.04 | 95.97 | 94.30 |
| EDL[54] | 11.56 | 88.60 | 93.92 | 19.95 | 99.07 | 95.70 | 19.36 | 93.15 | **96.54** | 16.96 | 93.61 | 95.39 | 95.72 |
| RED[49] | 65.75 | 29.85 | 61.30 | 86.49 | 71.56 | 28.06 | 72.37 | 19.83 | 51.16 | 74.87 | 40.41 | 46.84 | 95.80 |
| HEDL(Ours) | 8.43 | 94.09 | 96.86 | **19.15** | **99.19** | **96.23** | **19.08** | 90.14 | 95.71 | **15.55** | **94.47** | **96.27** | 95.66 |
| **CIFAR-100** | | | | | | | | | | | | | |
| MSP[16] | 83.69 | 60.76 | 76.04 | 83.83 | 85.24 | 76.93 | 81.24 | 62.39 | 79.44 | 82.91 | 69.46 | 77.47 | 77.25 |
| ODIN[31] | 89.76 | 52.36 | 71.08 | 78.37 | 86.67 | 79.39 | 81.27 | 60.85 | 79.83 | 83.13 | 66.62 | 76.77 | 77.25 |
| openGAN[26] | 83.96 | 60.85 | 78.68 | 86.31 | 80.18 | 73.53 | 88.37 | 38.87 | 70.15 | 86.21 | 59.96 | 74.12 | 77.25 |
| GradNorm[21] | 69.90 | 89.45 | 76.95 | 92.51 | 56.77 | 64.58 | 95.32 | 88.78 | 69.69 | 85.91 | 78.33 | 70.41 | 77.25 |
| VIM[66] | 82.79 | 72.82 | 81.20 | **55.90** | 92.15 | 87.41 | 83.85 | 56.24 | 75.76 | 74.18 | 73.74 | 81.46 | 77.25 |
| KNN[61] | 74.27 | 71.46 | 82.21 | 66.40 | 89.44 | 83.81 | 78.74 | 57.47 | 79.10 | 73.13 | 72.79 | 81.71 | 77.25 |
| DICE[59] | 79.93 | 65.95 | 79.97 | 80.53 | 85.41 | 77.70 | 80.75 | 62.76 | 80.18 | 80.40 | 71.37 | 79.28 | 77.25 |
| RankFeat[58] | 58.49 | 83.40 | 72.14 | 66.87 | 52.42 | 69.40 | 77.42 | 83.74 | 63.82 | 67.59 | 73.19 | 68.45 | 77.25 |
| ASH[8] | 46.00 | **92.97** | 85.60 | 61.27 | 68.97 | 80.72 | 62.95 | 91.48 | 81.69 | 56.74 | 84.47 | 81.69 | 77.25 |
| SHE[71] | 59.15 | 90.85 | 80.97 | 73.29 | 60.87 | 73.64 | 65.24 | 90.31 | 76.30 | 65.89 | 80.68 | 76.97 | 77.25 |
| GEN[38] | 55.45 | 90.36 | 81.41 | 61.23 | 64.52 | 78.74 | **56.25** | **91.90** | 80.28 | 57.64 | 82.26 | 80.14 | 77.25 |
| MCDropout[12] | 71.63 | 67.44 | 81.31 | 80.16 | 86.01 | 77.93 | 79.52 | 61.34 | 79.20 | 77.11 | 71.60 | 79.48 | 75.83 |
| G-ODIN[19] | 71.62 | 79.80 | 86.13 | 58.01 | 93.01 | **88.35** | 78.67 | 55.45 | 78.15 | 69.44 | 76.09 | 84.21 | 74.46 |
| CSI[62] | 67.21 | 91.76 | 80.24 | 90.51 | 51.46 | 62.22 | 69.41 | 88.16 | 70.99 | 75.71 | 77.13 | 71.15 | 61.60 |
| MOS[20] | 90.58 | 74.48 | 59.42 | 96.32 | 89.60 | 46.69 | 92.64 | 71.87 | 68.21 | 93.18 | 78.64 | 55.69 | 76.98 |
| VOS[9] | 98.62 | 56.36 | 68.99 | 94.54 | 76.20 | 68.33 | 97.81 | 43.20 | 68.21 | 96.99 | 58.59 | 68.51 | 77.20 |
| LogitNorm[67] | 79.16 | 75.57 | 83.03 | 87.06 | 79.08 | 71.53 | 80.20 | 63.10 | 79.84 | 82.14 | 72.58 | 78.13 | 76.34 |
| EDL[54] | 93.05 | 75.48 | 81.39 | 95.48 | 93.80 | 71.60 | 99.30 | 68.57 | 76.55 | 95.94 | 79.28 | 76.51 | 71.40 |
| RED[49] | 90.09 | 62.75 | 76.41 | 56.01 | 96.25 | 85.29 | 68.11 | 64.75 | 84.46 | 71.40 | 74.58 | 82.05 | 80.36 |
| HEDL(Ours) | **39.56** | 89.22 | **93.46** | 61.97 | **96.85** | 85.98 | 63.89 | 81.14 | **89.32** | **55.14** | **89.07** | **89.59** | **80.40** |

**Implementation Details.** We follow the experiment settings outlined in OpenOOD[69]. We use ResNet-18[15] for CIFAR-10 and CIFAR-100. For more intricate datasets that are not included in OpenOOD[69], such as fine-grained datasets Flower-102 and CUB-200-2011, we employ ResNet-34[15] for enhanced feature representation. All experiments are implemented with PyTorch[50] and carried out with NVIDIA GeForce RTX 3090 GPU. We use the standard data split for all datasets, and the number of training epochs is 100, the initial learning rate is 0.0001 with AdamW[39], and the batch size is 128. At test time, all images are resized to $224 \times 224$. For HEDL model, we first train the feature extractor with softmax layer for 90 epochs and then train in HEDL framework for 10 epochs. HEDL does not introduce any additional hyperparameters, thereby eliminating the need for extensive hyperparameter tuning, and $W_{prior}$ is set to 1 for HEDL.

**Baseline Methods.** We compare our method with several classical and state-of-the-art OOD detection methods. Specifically, we compare our method with post-hoc inference methods and training methods. From MSP[16] to GEN[38] are post-hoc inference methods, which affect OOD detection performance only and do not change model accuracy. The others are training methods. We excluded methods that required auxiliary OOD data due to the practical real-world situations consideration. We leverage selected experimental results from OpenOOD[69] to demonstrate the effectiveness of our approach.

Table 2: Ablation experiment results on Flower-102 and CUB-200-2011. Results show that EDL fails to extract evidence fully. HEDL without projection can extract comprehensive evidence to distinguish ID and OOD samples but fails to classify ID categories. HEDL can further assign evidence correctly and obtain accurate classification.

| | | | Flower-102 | | | | CUB-200-2011 | | | |
| | | | Average OOD performance | | | ID data | Average OOD performance | | | ID data |
| Multinomial-opinion | Hyper-opinion | Opinion-projection | FPR95↓ | AUPR↑ | AUROC↑ | Acc.↑ | FPR95↓ | AUPR↑ | AUROC↑ | Acc.↑ |
|---|---|---|---|---|---|---|---|---|---|---|
| - | - | - | 14.86 | 95.94 | 97.42 | 83.75 | 30.29 | 91.18 | 94.35 | **75.82** |
| ✓ | - | - | 100.00 | 66.95 | 67.23 | 66.84 | 98.03 | 71.80 | 75.27 | 59.87 |
| ✓ | ✓ | - | 11.90 | 95.83 | 97.61 | 81.40 | 9.32 | 91.57 | 97.82 | 52.30 |
| ✓ | ✓ | ✓ | **3.98** | **98.73** | **99.07** | **84.13** | **3.82** | **97.80** | **98.91** | 74.62 |

## 4.2 OOD Detection Results

The comparative results on CIFAR-10 and CIFAR-100 are detailed in Table 1, and the results on Flower-102 and CUB-200-2011 are shown in Appendix C. For each model, we utilize three OOD datasets, thereby aiming to achieve more realistic and generalized outcomes. We reveal a common challenge: when confronted with more complex data scenarios, training methods struggle to maintain both accuracy and OOD detection capabilities simultaneously. However, HEDL consistently achieves better OOD detection performance than existing state-of-the-art OOD detection methods while preserving the accuracy of ID classification, even under complex data scenarios. Notably, HEDL accomplishes this enhancement without additional regularization strategies or hyperparameters, indicating strong generalization ability on different datasets, it also avoids incurring higher computational costs. The experimental training time analysis of HEDL can be found in Appendix D.

## 4.3 Gradient Analysis

The gradient norms of fully-connected layer parameters over EDL and HEDL during training is shown in Figure 3, alongside the final accuracy for each category. The sum of these gradient norms has been normalized for comparative analysis. It is observed that the gradient norms for several parameters within the fully-connected layer of the EDL model remain zero throughout the training process, which correlates with a significantly lower final accuracy for these categories. This outcome is indicative of the vanishing gradient problem. Conversely, HEDL does not experience this issue, demonstrating that our proposed method effectively circumvents the challenge of vanishing gradients within the fully-connected layer.

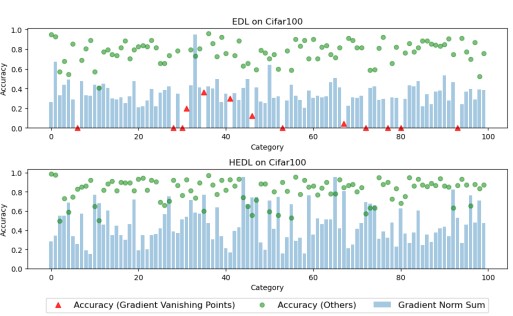

Figure 3: The sum of gradient norms within the fully-connected layer for each category in CIFAR-100 throughout the training process.

## 4.4 Ablation Study

EDL suffers from a notable decline in both ID accuracy and OOD detection when facing a proportional rise in the volume of vague evidence. In contrast, HEDL demonstrates the capability to consistently extract comprehensive evidence and maintain its performance regardless of the dataset scale.

We investigate the performance of our method with ablation experiments on two challenging fine-grained datasets. The fine-grained datasets contain more vagueness among categories and can better prove the effectiveness of our methods. We conducte ablation experiments on the effects of hyper-opinion and opinion projection, respectively. Note that opinion projection can only be built upon hyper-opinion.

Figure 4 illustrates the uncertainty distribution of ID and OOD samples across different datasets for EDL, HEDL without opinion projection, and HEDL itself. Notably, on the latter three more complex datasets, the approaches based on hyper-opinion exhibits a distinct performance advantage. It is also

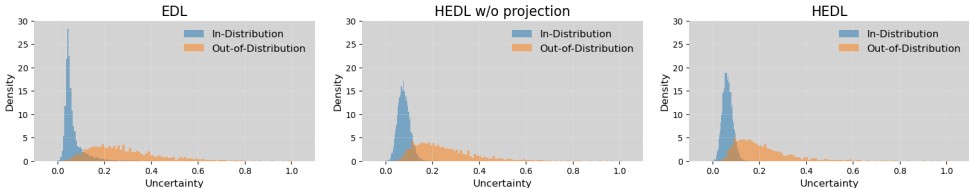

(a) CIFAR-10, the overlap between ID and OOD is 20%, 23%, and 18% for EDL, HEDL w/o projection, and HEDL, respectively.

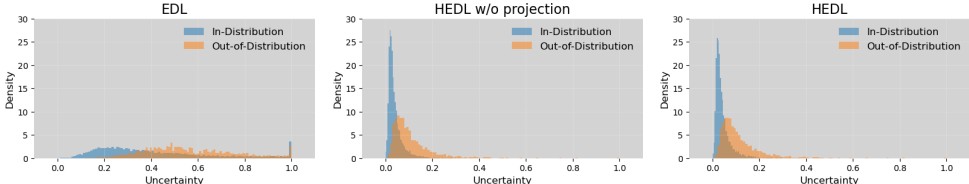

(b) CIFAR-100, the overlap between ID and OOD is 62%, 45%, and 41% for EDL, HEDL w/o projection, and HEDL, respectively.

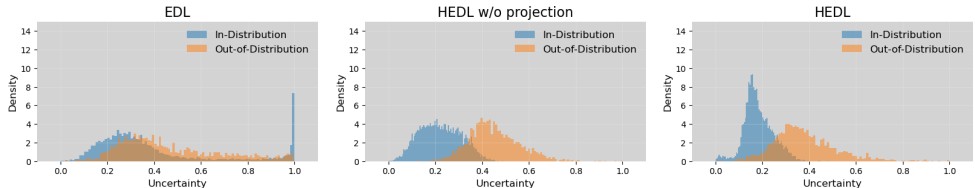

(c) Flower-102, the overlap between ID and OOD is 71%, 26%, and 29% for EDL, HEDL w/o projection, and HEDL, respectively.

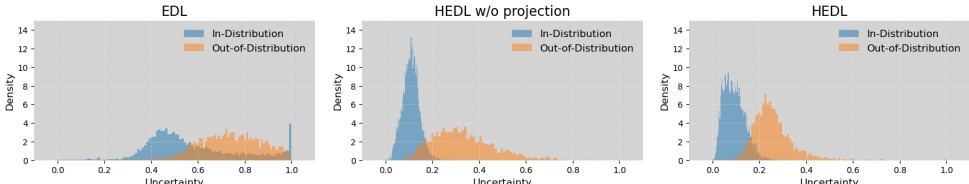

(d) CUB-200-2011, the overlap between ID and OOD is 50%, 20%, and 17% for EDL, HEDL w/o projection, and HEDL, respectively.

Figure 4: The normalized density distribution of normalized uncertainty for ID and OOD samples across differing datasets.

worth observing that, in these datasets, instances of ID data with maximum uncertainty are present in the EDL model. This phenomenon can be attributed to the failure of extracting evidence of those categories due to the vanishing gradient problem.

Table 2 shows that evidence built on hyper-opinion can be considered comprehensively, leading to accurate uncertainty estimation and above baseline OOD detection performance. But without the correct projection from hyper-opinion to multinomial-opinion, vague evidence can not be assigned precisely, leading to inaccurate classification.

## 5 Conclusion

In this paper, we propose Hyper-opinion Evidential Deep Learning (HEDL), a novel approach designed to generate precise uncertainty estimation for Out-of-Distribution (OOD) detection. Our method encapsulates a comprehensive representation of evidence within hyper-opinion, which allows model to preserve its vagueness among In-Distribution categories to reject OOD data.

Additionally, by projecting hyper-opinion to multinomial-opinion, HEDL circumvents the vanishing gradient problem encountered in the fully-connected layers of traditional EDL. This projection is optimized within an established framework, yielding accurate and reliable evidence. Notably, our method

accomplishes superior OOD detection performance while simultaneously upholding classification accuracy without incurring additional computational complexity. Extensive experimental results across numerous datasets substantiate the efficacy of the proposed Hyper-opinion Evidential Deep Learning.

**Limitations and societal impact.** Our proposed HEDL method achieves best performance by transfering learning on pre-trained models. In future work, it is necessary to reduce the dependence on pre-trained models and explore alternative approaches. This work aims to improve the safety of deep learning models, which tends to benefit a wide range of applications of AI in social life.

## Acknowledgments and Disclosure of Funding

This work was supported by the National Natural Science Foundation of China (No. 62173252, 62476165).

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

## A  An Example within EDL and HEDL

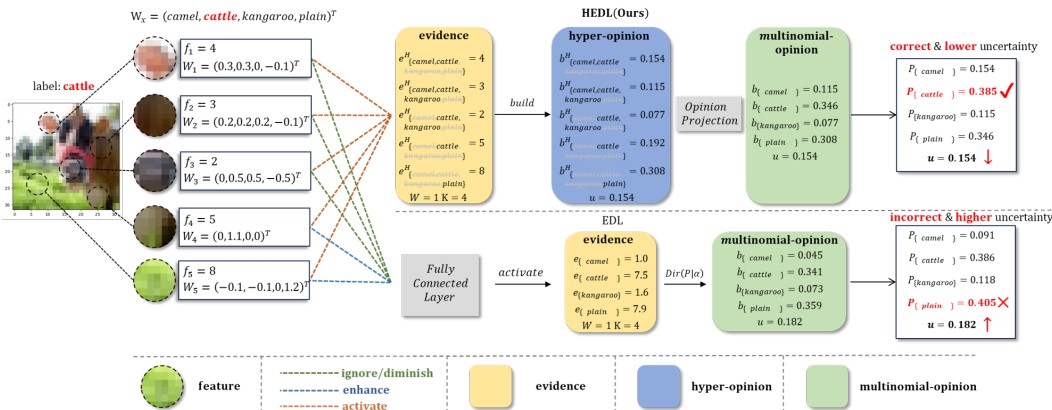

Figure 5: Example of an image been classified by EDL and HEDL. When confronted with vague samples, HEDL leverages the incorporation of vague evidence, which culminates in enhanced accuracy for classification and more precise uncertainty estimations, thereby fortifying OOD detection capabilities.

A sample displaying the classification process of EDL and HEDL is shown in Figure 5, EDL tends to ignore or diminish the amount of vague evidence to get sharper belief mass. The loss of evidence leads to increased uncertainty and potential misclassification. In contrast, HEDL framework reserves the vague evidence, thereby achieving improved estimations of uncertainty and more accurate classification results. Notice that when there is no evidence supporting a set $x$, then $e_x^H = 0$.

## B  Gradient Vanishing Analysis

**Proposition 1.** *By building evidence on hyper-opinion and then projecting to multinomial-opinion, we avoid the vanishing gradient problem in fully-connected layer in traditional EDL.*

**Proof 1.** Consider the neural network forward propagation in EDL

$$o_k = W\boldsymbol{z} + bias, \tag{18}$$

$$e_k = ReLU(o_k), \tag{19}$$

$$\alpha_k = e_k + \frac{W_{prior}}{K}, \tag{20}$$

$$\mathcal{L}_i(\Theta) = \sum_{j=1}^{K} y_{ij}\Big(\psi(S_i) - \psi(\alpha_{ij})\Big), \tag{21}$$

where $bias$ stands for the bias of the fully-connected layer, $\boldsymbol{z}$ represents the feature extracted by the neural network. We can write expressions for all partial derivatives as follows:

$$\frac{\partial o_k}{\partial W} = \boldsymbol{z}, \quad \frac{\partial \alpha_k}{\partial e_k} = 1, \tag{22}$$

$$\frac{\partial \mathcal{L}}{\partial \alpha_k} = \left( \frac{1}{S^2} + \sum_{i=1}^{\infty} \frac{1}{(i+S)^2} - \frac{y_k}{\alpha_{gt}^2} - \sum_{i=1}^{\infty} \frac{y_k}{(i+\alpha_{gt})^2} \right), \tag{23}$$

$$\frac{\partial e_k}{\partial o_k} = \begin{cases} 0 & \text{if} \quad o_k \leq 0 \\ 1 & \text{otherwise.} \end{cases} \tag{24}$$

Therefore by the chain rule, we can calculatethe the gradient *w.r.t.* $W$ as:

$$\frac{\partial \mathcal{L}}{\partial W} = \frac{\partial \mathcal{L}}{\partial \alpha_k}\frac{\partial \alpha_k}{\partial e_k}\frac{\partial e_k}{\partial o_k}\frac{\partial o_k}{\partial W} = \frac{\partial \mathcal{L}}{\partial \alpha_k}\frac{\partial e_k}{\partial o_k}\boldsymbol{z}, \tag{25}$$

Obviously when exists $o_k \leq 0, \forall k \in \mathbb{X}$, vanishing gradient problem is unavoidable in traditional EDL. To ensure that proposed HEDL is not associate with similar problem, considering the forward propagation of HEDL:

$$o_k = W\boldsymbol{z} + bias, \tag{26}$$

$$e_k = o_k G(W, \boldsymbol{b}^H), \tag{27}$$

$$\alpha_k = e_k + \frac{W_{prior}}{K}, \tag{28}$$

where $G(W, \boldsymbol{b}^H)$ can be calculated by Eq. 7 and Eq. 13, and the gradient *w.r.t.* $W$ is calculated by chain rule:

$$\frac{\partial \mathcal{L}}{\partial W} = \frac{\partial \mathcal{L}}{\partial \alpha_k} \frac{\partial \alpha_k}{\partial e_k} \frac{\partial e_k}{\partial o_k} \frac{\partial o_k}{\partial W}, \tag{29}$$

where $\frac{\partial \mathcal{L}}{\partial \alpha_k}, \frac{\partial \alpha_k}{\partial e_k}, \frac{\partial o_k}{\partial W}$ are known items that won't cause vanishing gradient problem. Consider

$$\frac{\partial e_k}{\partial o_k} = \frac{\partial o_k G(W, \boldsymbol{b}^H)}{\partial o_k} = G(W, \boldsymbol{b}^H), \tag{30}$$

where $W, \boldsymbol{b}^H$ are all detached variables that are irrelevant variables in this partial derivative item, implying that $G(W, \boldsymbol{b}^H)$ remains constant during the backward process.

$$\frac{\partial \mathcal{L}}{\partial W} = \frac{\partial \mathcal{L}}{\partial \alpha_k} G(W, \boldsymbol{b}^H)\boldsymbol{z}. \tag{31}$$

Consequently, the opinion projection successfully circumvents the vanishing gradient problem in the fully-connected layer.

## C   Experiment Results on Flower-102 and CUB-200-2011

Table 3 details the comparative results on two fine-grained datasets Flower-102 and CUB-200-2011. On more complex fine-grained datasets, HEDL consistently demonstrates superior performance in OOD detection.

Table 3: Comparison of OOD detection performance between HEDL and other baselines with Flower-102 and CUB-200-2011 as ID dataset.

| Method | Flower-102 | | | | CUB-200-2011 | | | |
| --- | --- | --- | --- | --- | --- | --- | --- | --- |
| | Average OOD performance | | | ID data | Average OOD performance | | | ID data |
| | FPR95↓ | AUPR↑ | AUROC↑ | Acc.↑ | FPR95↓ | AUPR↑ | AUROC↑ | Acc.↑ |
| MSP[16] | 14.86 | 95.94 | 97.42 | 83.75 | 30.29 | 91.18 | 94.35 | 75.82 |
| ODIN[31] | 4.36 | 97.63 | 98.22 | 83.75 | 21.92 | 89.92 | 96.22 | 75.82 |
| VIM[66] | 6.34 | 96.70 | 97.94 | 83.75 | 6.71 | 97.27 | 98.26 | 75.82 |
| GradNorm[21] | 5.38 | 97.11 | 98.81 | 83.75 | 32.08 | 97.68 | 95.22 | 75.82 |
| KNN[61] | 18.45 | 88.83 | 95.30 | 83.75 | 14.35 | 88.63 | 97.40 | 75.82 |
| DICE[59] | 4.64 | 97.62 | 98.95 | 83.75 | 25.82 | 88.83 | 96.00 | 75.82 |
| RankFeat[58] | 96.57 | 76.62 | 60.98 | 83.75 | 74.68 | 83.38 | 71.09 | 75.82 |
| ASH[8] | 5.16 | 97.54 | 98.84 | 83.75 | 15.82 | 92.75 | 97.07 | 75.82 |
| SHE[71] | 11.69 | 93.96 | 97.79 | 83.75 | 22.94 | 96.14 | 96.18 | 75.82 |
| GEN[38] | 5.25 | 97.55 | 98.85 | 83.75 | 15.88 | 92.74 | 97.06 | 75.82 |
| MCDropout[12] | 14.77 | 96.22 | 97.41 | 83.98 | 42.46 | 87.08 | 91.76 | 75.83 |
| G-ODIN[19] | 56.92 | 69.88 | 82.12 | 24.30 | 29.51 | 85.13 | 93.85 | 66.74 |
| VOS[9] | 39.17 | 84.52 | 90.11 | 78.08 | 35.98 | 83.93 | 89.86 | 75.92 |
| LogitNorm[67] | 41.07 | 80.34 | 85.65 | 77.41 | 22.69 | 91.69 | 95.99 | 74.84 |
| EDL[54] | 100.00 | 66.95 | 67.23 | 66.84 | 98.03 | 71.80 | 75.27 | 59.87 |
| RED[49] | 95.87 | 80.10 | 76.45 | **84.63** | 36.01 | 94.58 | 94.89 | **76.30** |
| HEDL(Ours) | **3.98** | **98.73** | **99.07** | 84.13 | **3.82** | **97.80** | **98.91** | 74.62 |

# D Experiment Analysis of Computational Complexity

Table 4 presents the average training time per epoch of EDL and HEDL compared with MSP on different datasets, all under identical training conditions. The results indicate that the implementation of HEDL does not incur additional computational complexity.

Table 4: Average training time per epoch of EDL and HEDL compared with MSP on different datasets, + indicates more time, and - indicates less time.

| Method | Cifar10 | Cifar100 | Flower-102 | CUB-200-2011 |
|--------|---------|----------|------------|--------------|
| EDL    | +3.78%  | -1.93%   | +1.21%     | +2.14%       |
| HEDL   | +1.62%  | +1.02%   | -0.74%     | +3.57%       |

