# OpenReview forum: "Hyper-opinion Evidential Deep Learning for Out-of-Distribution Detection"
_NeurIPS.cc/2024/Conference — NeurIPS 2024 poster_

### Official Review · Reviewer_cb7H · 2024-07-01

**Soundness:** 4
**Presentation:** 3
**Contribution:** 4
**Rating:** 7
**Confidence:** 4

**Summary:**

This manuscript takes a significant step forward in the realm of evidential deep learning by considering a more holistic hyper-opinion evidence framework.  The approach presented offers a novel perspective for optimizing evidential deep learning models, notably enhancing their ability to detect out-of-distribution instances without additional settings. This enhancement is particularly evident when the models are applied to complex datasets.

**Strengths:**

The manuscript is commendable for presenting a fresh and innovative approach to the framework of evidential deep learning. It is noteworthy that while previous research has mainly concentrated on enhancing the settings in evidential deep learning, there has been an oversight in addressing the structural deficiencies of the framework itself.

The authors have introduced an interesting concept by treating the features extracted by the network as evidence when applying the theory to practical scenarios. The authors also give full mathematical proof on the problems they solved. This approach grounds in a solid theoretical foundation based on subjective logic, demonstrates simplicity in its application and shows impressive results in experimental validation.

**Weaknesses:**

The manuscript exhibits some shortcomings in the handling of certain details. The hyper-domain does not encompass the set itself and the empty set; however, in practice, due to the treatment of features as evidence, the possibility of evidence representing these two sets cannot be ruled out. Furthermore, the subjective logic stipulates that evidence from the same subset should appear at most once, yet the approach presented in this paper does not guarantee full compliance with this condition.

**Questions:**

I had some concerns on the setup of the ablation experiment. What implement the authors take to transform from hyper-opinion to multinomial-opinion without opinion projection?

**Limitations:**

The limitation is understandable and the method does not present any potential negative societal impact.

---

> ### Author Rebuttal · Authors · 2024-08-06
>
> Dear Reviewer cb7H,
>
> We sincerely appreciate the time and effort you have dedicated to reviewing our paper and providing us with valuable feedback. Here are our replies.
>
> > The manuscript exhibits some shortcomings in the handling of certain details. The hyper-domain does not encompass the set itself and the empty set; however, in practice, due to the treatment of features as evidence, the possibility of evidence representing these two sets cannot be ruled out. Furthermore, the subjective logic stipulates that evidence from the same subset should appear at most once, yet the approach presented in this paper does not guarantee full compliance with this condition.
>
> Due to the opinion projection without prior information, both the full set and the empty set contribute the same to each category. For full set, it contributes $\frac{1}{K}$ (where K is the number of categories) evidence to each category. For empty set, it contributes none evidence to any category. Therefore both situation will not affect the model accuracy.
>
> Before the opinion projection, there may exists multiple pieces of evidence supporting the same set. But these evidences will be accumulated as one piece of evidence, which meets the condition of Subjective Logic. As in the 'Hyper-opinion Belief' section of Figure 2, the belief mass for one set appears at most once.
>
> > I had some concerns on the setup of the ablation experiment. What implement the authors take to transform from hyper-opinion to multinomial-opinion without opinion projection?
>
> We first model the evidence on hyper-domain to form the hyper-opinion, and then use the fully connected layer for opinion assignment. ReLU activation is applied for the fully connected layer weights and the bias is set to 0 to ensure the non-negative assignment of hyper-opinion evidence.

---

### Official Review · Reviewer_Y3V6 · 2024-07-03

**Soundness:** 4
**Presentation:** 3
**Contribution:** 3
**Rating:** 6
**Confidence:** 4

**Summary:**

This paper studies the problem of out-of-distribution (OOD) detection. Traditional Evidential Deep Learning framework collects sharp evidence that supports a single category while ignoring vague evidence that supports multiple categories, leading to inaccurate uncertainty estimation and decreased OOD detection performance. The authors introduce hyper-domain and propose Hyper-opinion Evidential Deep Learning. With hyper-opinion, HEDL explicitly models evidences as sharp and vague evidences that support single categories and multiple categories respectively. HEDL extends the framework of EDL and establishes more accurate uncertainty estimation for OOD detection. Experiments on several datasets demonstrate the effectiveness of the proposed method.

**Strengths:**

-The proposed method is sound in theory. HEDL extends the framework of EDL, establishes a more accurate uncertainty estimation framework for OOD detection. It’s quite novel and enlightening.
-Without additional regularization terms nor computational complexity, it maintains the simplicity and generalizability.
-This paper has sufficient experiments to demonstrate the effectiveness of the proposed method and each module. It achieves better performance compared with SOTA methods.

**Weaknesses:**

-To ensure fairness, the value of W_{prior} should be explicitly stated and consistent with EDL, cause the value of W_{prior} has been proved to be crucial to the effectiveness of EDL model [1].
-The authors may consider incorporating the KL divergence as a loss term to enhance the model's performance.
-In the section when proofing the existence of the gradient vanishing problem in EDL. It would be beneficial to delineate the conditions where o_k is less than zero occurs.
-The paper could benefit from a wider analysis of the impact of the other two loss functions mentioned in EDL, e.g., the MSE loss and the Log loss, when applied to the HEDL model.

[1] Chen M, Gao J, Xu C. R-EDL: Relaxing Nonessential Settings of Evidential Deep Learning[C]//The Twelfth International Conference on Learning Representations. 2023.

**Questions:**

1.Whether the opinion projection process that translates hyper-opinion into standard multinomial-opinion, leads to a loss of uncertainty information? Please provide a detailed discussion on this matter.
2.Whether HEDL encounters the issue of exponential explosion, a challenge mentioned in subjective logic as the number of categories expands. Could the authors address this potential concern and discuss any mechanisms to manage such an increase in complexity?

**Limitations:**

The limitations discussed in the manuscript do not appear to significantly constrain the applicability of the method.

---

> ### Author Rebuttal · Authors · 2024-08-06
>
> Dear Reviewer Y3V6,
>
> We sincerely appreciate the time and effort you have dedicated to reviewing our paper and providing us with valuable feedback. Here are our replies.
>
> > To ensure fairness, the value of $W_{prior}$ should be explicitly stated and consistent with EDL, cause the value of $W_{prior}$ has been proved to be crucial to the effectiveness of EDL model [1].
>
> In all our experiments, the value of $W_{prior}$ is set to 1, the same as in EDL, denoting no prior information of the evidence.
>
> This clarification is mentioned in section 3.1, if the paper is accepted, we will add this definition to the experimental setup too.
>
> > The authors may consider incorporating the KL divergence as a loss term to enhance the model's performance.
>
> We tried to add KL divergence as a constraint yet obtained worse model effect. In fact, HEDL will extract and retain vague evidence, while KL divergence tends to constrain the generation of vague evidence. The conflict between them thus affecting the effect of the model.
>
> > In the section when proofing the existence of the gradient vanishing problem in EDL. It would be beneficial to delineate the conditions where $o_k$ is less than zero occurs.
>
> Eq. 27 shows that in each back-propagation process, for categories that are not ground-truth, the gradient values of the fully connected layer weights will experience large gradient descent. For some categories (determined by the initial parameters of the model), after undergoing multiple large gradient descents, the fully connected layer weights are degraded, resulting in the output $o_k$ being less than 0.
>
> > The paper could benefit from a wider analysis of the impact of the other two loss functions mentioned in EDL, e.g., the MSE loss and the Log loss, when applied to the HEDL model.
>
> Thanks for your suggestion. The results of applying the other two loss functions are as follows. Different loss functions have limited effect on the model performance.
>
> | Method | CIFAR-10 |     |     |     | CIFAR-100 |     |     |     |
> | --- | --- | --- | --- | --- | --- | --- | --- | --- |
> |     | FPR95 | AUPR | AUROC | Acc | FPR95 | AUPR | AUROC | Acc |
> | MSE | 18.27 | 92.66 | 95.16 | 95.47 | 56.14 | 89.15 | 88.65 | 80.67 |
> | Log | 15.33 | 93.64 | 95.71 | 95.62 | 52.44 | 89.70 | 89.15 | 80.39 |
> | Digamma | 15.55 | 94.47 | 96.27 | 95.66 | 55.14 | 89.07 | 89.59 | 80.40 |
>
> > Whether the opinion projection process that translates hyper-opinion into standard multinomial-opinion, leads to a loss of uncertainty information? Please provide a detailed discussion on this matter.
>
> In the process of projecting hyper-opinion to multinomial-opinion, vague evidence is allocated to specific singletons, while the total evidence mass does not change. At the same time, the uncertainty of each singleton and the overall uncertainty remains unchanged. Due to Eqs. 10 and 18, there is no uncertainty loss in the opinion projection process.
>
> > Whether HEDL encounters the issue of exponential explosion, a challenge mentioned in subjective logic as the number of categories expands. Could the authors address this potential concern and discuss any mechanisms to manage such an increase in complexity?
>
> Instead of building the corresponding evidence for each element, we build the extracted features as evidence in the hyper-domain. The number of evidence we extract are determined by the feature dimensions. So the number of evidence does not change with the number of categories, which avoids the exponential explosion problem.

---

> > ### Comment · Reviewer_Y3V6 · 2024-08-13
> > **Retaining my positive score**
> >
> > Thank you for your rebuttal. I am glad that the authors validated the wider analysis I raised for KL divergence and other loss functions. The definition of W_{prior} addressed my concern about the fairness, and please include the definition in the paper. The discussion on uncertainty loss and exponential explosion is convincing. After reading other reviews and your responses, I am going to retain my score.

---

> > > ### Author Response · Authors · 2024-08-13
> > >
> > > Thank you for helping us improve the paper, we will add the definition about $W_{prior} $ to the experimental setup in an updated version of this paper. We really appreciate your valuable comment!

---

### Official Review · Reviewer_jaEe · 2024-07-09

**Soundness:** 3
**Presentation:** 3
**Contribution:** 3
**Rating:** 6
**Confidence:** 4

**Summary:**

This paper introduces an out-of-distribution detection method based on evidential deep learning. This method models the evidence in hyper-domain, and the hyper-opinion in Subjective Logic is used to replace the multinomial-opinion in traditional evidential deep learning. The Hyper-opinion Evidential Deep Learning considers the vague evidence ignored in traditional Evidential Deep Learning, so as to achieve better uncertainty estimation effect. The OOD detection performance of proposed method exceeds the current SOTA OOD detection methods while maintaining the classification accuracy.

**Strengths:**

1.	HEDL achieves SOTA OOD detection performance, while maintaining classification accuracy.
2.	As the number of categories in a dataset increases, the traditional EDL framework's performance significantly deteriorates. In contrast, HEDL can consistently extract comprehensive evidence and maintain its performance, regardless of the dataset's scale.
3.	HEDL does not introduce additional computational complexity of the model.
4.	HEDL can mitigate the vanishing gradient problem in EDL theoretically and practically.

**Weaknesses:**

1.	The origin of the sample uncertainty depicted in the upper portion of Figure 1 is not clearly defined. It is imperative to clarify whether the data presented comes from actual experimental outcomes or if it is merely illustrative example. If the data is based on real-world results, please provide an explanation of how to quantify the samples vagueness.
2.	The paper does not specify the value of $W_{prior} $ nor does it analyze its role as a hyperparameter. It is essential to elucidate whether $W_{prior} $ is equivalent to EDL. Please provide a detailed numerical definition of $W_{prior} $.
3.	Figure 2 can be described more clearly. In the second module ‘Opinion Projection’ of the lower part, the positioning of the right bar appears to be more logically associated with the 'Multinomial-Opinion Optimization' section. To improve the figure's comprehensibility and accuracy, it is recommended to reposition the right bar accordingly.

**Questions:**

1.	In Figure 4, the uncertainty distribution of In-Distribution and Out-of-Distribution data is similar between HEDL w/o projection and HEDL. This observation raises the question of the contribution of the opinion projection. Could the authors further explain the specific role and impact of the opinion projection in the whole method?
2.	Could the authors clarify the conceptual difference between a 'Dirichlet hyper distribution' and a standard 'Dirichlet distribution' in Equation 8? From my understanding, it seems that the two distributions equal. If there is a nuanced difference, please provide an explanation to delineate the unique characteristics of the 'Dirichlet hyper distribution' as employed in your method.

**Limitations:**

Yes, limitations and social impact are discussed in the main paper.

---

> ### Author Rebuttal · Authors · 2024-08-06
>
> Dear Reviewer jaEe,
>
> We sincerely appreciate the time and effort you have dedicated to reviewing our paper and providing us with valuable feedback. Here are our replies.
>
> > The origin of the sample uncertainty depicted in the upper portion of Figure 1 is not clearly defined. It is imperative to clarify whether the data presented comes from actual experimental outcomes or if it is merely illustrative example. If the data is based on real-world results, please provide an explanation of how to quantify the samples vagueness.
>
> These data comes from the experimental results of CIFAR-100. We randomly select two categories (kangaroos and dinosaurs in Figure 1 for example), and calculate the vague evidence ratio of each sample in HEDL. Next we calculate the uncertainty of the samples in EDL and HEDL models respectively.
>
> > The paper does not specify the value of $W_{prior}$ nor does it analyze its role as a hyperparameter. It is essential to elucidate whether $W_{prior}$ is equivalent to EDL. Please provide a detailed numerical definition of $W_{prior}$.
>
> In all our experiments, the value of $W_{prior}$ is set to 1, the same as in EDL, denoting no prior information of the evidence.
>
> This clarification is mentioned in section 3.1, if the paper is accepted, we will add this definition to the experimental setup too.
>
> > Figure 2 can be described more clearly. In the second module ‘Opinion Projection’ of the lower part, the positioning of the right bar appears to be more logically associated with the 'Multinomial-Opinion Optimization' section. To improve the figure's comprehensibility and accuracy, it is recommended to reposition the right bar accordingly.
>
> Thanks for your suggestion. The 'Multinomial-Opinion Optimization' section is about the optimization of the Dirichlet distribution associates with a multinomial-opinion. The right bar in ‘Opinion Projection’ represents the belief mass allocation from hyper-opinion to multinomial-opinion, which is more related to opinion projection. If the paper is accepted, we will make Figure 2 more clear.
>
> > In Figure 4, the uncertainty distribution of In-Distribution and Out-of-Distribution data is similar between HEDL w/o projection and HEDL. This observation raises the question of the contribution of the opinion projection. Could the authors further explain the specific role and impact of the opinion projection in the whole method?
>
> As shown in **Table 2**, the evidence needs to be projected from hyper-opinion to multinomial-opinion for accurate classification. The opinion projection operation can ensure the correct allocation of vague and sharp evidence, thereby obtaining accurate classification results.
>
> > Could the authors clarify the conceptual difference between a 'Dirichlet hyper distribution' and a standard 'Dirichlet distribution' in Equation 8? From my understanding, it seems that the two distributions equal. If there is a nuanced difference, please provide an explanation to delineate the unique characteristics of the 'Dirichlet hyper distribution' as employed in your method.
>
> The Dirichlet hyper distribution is built upon hyper-domain.
> Instead of building evidence for singletons in a domain, Dirichlet hyper distribution treats the set as a singleton element in hyper-domain.
> Thus Dirichlet hyper distribution is able to model the evidence for set (support multiple singletons in the same time), while Dirichlet distribution is limited to model evidence for singletons only. Modeling evidence for set enables Dirichlet hyper distribution to take vague evidence into consideration.

---

### Official Review · Reviewer_hxrE · 2024-07-10

**Soundness:** 3
**Presentation:** 2
**Contribution:** 3
**Rating:** 6
**Confidence:** 3

**Summary:**

This paper provides a method for Out-of-Distribution detection called Hyper-opinion Evidential Deep Learning which is based on Evidential Deep Learning. It models the evidence on the hyper-domain, considering the extra vague evidence containing multiple possible categories. The measurement of vague evidence makes HEDL have more accurate uncertainty estimation. This paper also proposes an opinion projection method to mitigate the gradient vanishing problem in EDL. Experiments on several datasets show that the proposed method outperforms the current Out-of-Distribution detection methods.

**Strengths:**

(1)The paper has clear motivation for each part of the proposed method and the method has a solid theoretical foundation.

(2)The proposed method achieves SOTA results compared with other methods.

**Weaknesses:**

1.Eqs. 13, 14, and 15 are hard to understand. The authors can detail on the meanings between the different W's.

2.Why HEDL is trained on a pre-trained feature extractor and how to ensure the fairness of comparative evaluations with other methods?

3.Theoretical proof of how HEDL solves vanishing gradient problem is missing. How is the EDL gradient vanishing point determined in Figure 3 and does this phenomenon consistently result in the vanishing gradients for certain fixed classes during each iteration?

4.Adding experiments on model complexity comparison can be beneficial to illustrate the computation complexity of HEDL.

5.More state-of-the-art methods (CIDER[1] and NECO[2]) can better demonstrate the effectiveness of proposed method.

[1] https://openreview.net/forum?id=aEFaE0W5pAd

[2] https://openreview.net/forum?id=9ROuKblmi7

 I’m mostly concerned about 2 and 3. If the authors can address these questions, I may consider raising my score.

**Questions:**

Please answer the questions in Weakness.

**Limitations:**

Please see Weakness

---

> ### Author Rebuttal · Authors · 2024-08-06
>
> Dear Reviewer hxrE,
>
> We sincerely appreciate the time and effort you have dedicated to reviewing our paper and providing us with valuable feedback. Here are our replies.
>
> > Eqs. 13, 14, and 15 are hard to understand. The authors can detail on the meanings between the different W's.
>
> $W$ is a matrix of $N \times K$ shape, representing the weight of the fully connected layer.
>
> $W^s$ is a 0/1 matrix obtained by applying the Heaviside function to each value in the $W$ matrix, representing the set of each piece of evidence supports on the hyper-domain.
>
> $W^p$ normalizes the value of each row of $W^s$ according to prior information, representing the opinion projection weight matrix.
>
> > Why HEDL is trained on a pre-trained feature extractor and how to ensure the fairness of comparative evaluations with other methods?
>
> Because HEDL extracts both vague and sharp evidences, it converges slowly in the early stages of training. Therefore, applying the softmax layer to pre-train a feature extractor and then fine-tuning it with HEDL can achieve faster convergence and better model results.
>
> Refer to **Experiment Implementation Details**. For all comparative experiments, we apply the same model structure and initial training parameters, as well as the total number of training epochs, thus ensuring the fairness.
>
> > Theoretical proof of how HEDL solves vanishing gradient problem is missing. How is the EDL gradient vanishing point determined in Figure 3 and does this phenomenon consistently result in the vanishing gradients for certain fixed classes during each iteration?
>
> We provided theoretical proof of how HEDL solves vanishing gradient problem in **Appendix B**.
>
> Figure 3 shows that in each training process, the parameters whose gradient norm sum is 0 are the gradient vanishes points.
> The gradient vanishing classes in each training are not fixed, and will change according to the initial parameters.
> Due to lacking of extracting evidence training caused by gradient vanishing, those gradient vanishing classes suffer from low classification accuracy and high uncertainty.
>
> > Adding experiments on model complexity comparison can be beneficial to illustrate the computation complexity of HEDL.
>
> We had provided the experimental data with training time for each model(softmax/EDL/HEDL) in **Appendix D**. Both theoretical proof and experimental results show that HEDL does not increase the computational complexity of the model.
>
> > More state-of-the-art methods (CIDER[1] and NECO[2]) can better demonstrate the effectiveness of proposed method.
>
> Thanks for your suggestion. We planned to compare with CIDER in our experiments, since CIDER does not report the classification accuracy in its original paper, we did not put it the main text. As for NECO, we keep the same experimental settings and model parameters as in our paper to reproduce the experimental results. The results are shown in the table below. It proves that our method is better than the current SOTA methods. If our paper is accepted, we can add these experimental results to our paper if necessary.
>
> | Method | CIFAR-10  |           |           | CIFAR-100 |           |           |
> |:------ |:---------:|:---------:|:---------:|:---------:|:---------:|:---------:|
> |        | FPR95     | AUPR      | AUROC     | FPR95     | AUPR      | AUROC     |
> | CIDER  | 21.94     | **95.16** | 95.33     | 58.21     | 87.99     | 84.61     |
> | NECO   | 31.28     | 92.34     | 94.68     | 64.92     | 86.36     | 83.74     |
> | HEDL   | **15.55** | 94.47     | **96.27** | **55.14** | **89.07** | **89.59** |

---

> > ### Comment · Reviewer_hxrE · 2024-08-11
> >
> > I would like to thank the authors for the additional explanation and more experimental results. I look forwards to seeing an updated version of this paper with these results, and I have updated my score.

---

> > > ### Author Response · Authors · 2024-08-12
> > >
> > > Thank you for helping us improve the paper and updating the score, we really appreciate your valuable comment!

---

### Decision · Program_Chairs · 2024-09-25

**Decision:**

Accept (poster)

**Comment:**

The paper introduces Hyper-opinion Evidential Deep Learning (HEDL), an extension of the Evidential Deep Learning (EDL) framework, for out-of-distribution detection. The method addresses the limitations of EDL by incorporating both sharp evidence (supporting single categories) and vague evidence (supporting multiple categories). The authors introduce an opinion projection mechanism that mitigates the vanishing gradient issue encountered in EDL. The method is shown to outperform existing OOD detection methods across various datasets.

All reviewers acknowledged its significant contributions, sound theoretical foundation, and solid experimental validation. They consistently rated the paper positively, with all ratings falling within the weak accept-to-accept range.

The main strengths of the paper:
The paper presents a novel extension to the EDL framework, enhancing its capability for OOD detection by effectively modeling both sharp and vague evidence. The introduction of the opinion projection mechanism is a meaningful contribution, addressing the vanishing gradient issue and improving classification accuracy without increasing model complexity. Extensive experiments demonstrate that HEDL outperforms state-of-the-art methods in OOD detection, confirming the practical relevance of the proposed approach.
The paper is grounded in solid theoretical foundations and includes rigorous mathematical proofs to support the proposed innovations.

Weaknesses:
Some equations were initially difficult to understand, and the paper could benefit from additional clarifications and improvements in the presentation of certain concepts before the camera ready. Concerns were raised about the fairness of comparative evaluations due to the use of a pre-trained feature extractor. The authors clarified that consistent experimental conditions were applied across all methods.
Minor concerns were noted about the clarity of some figures and the explanation of certain terms, which were addressed in the author rebuttal.

Discussion: The authors effectively addressed all concerns raised by the reviewers during the rebuttal process. They provided additional explanations, clarified ambiguous points, and presented further experimental results, which satisfied the reviewers and led to an overall positive reception of the paper.

All in all, given the strong theoretical contributions, robust experimental validation, and positive feedback from reviewers, I recommend accepting this paper. The paper is expected to have a positive impact in the area of probabilistic methods and OOD detection, contributing valuable insights to the NeurIPS community.